# AT YOUR FINGERTIPS:
# AUTOMATIC PIANO FINGERING DETECTION

## ABSTRACT

Automatic Piano Fingering is a hard task which computers can learn using data. As data collection is hard and expensive, we propose to automate this process by automatically extracting fingerings from public videos and MIDI files, using computer-vision techniques. Running this process on 90 videos results in the largest dataset for PIANO-FINGERING with more than 150K notes. We show that when running a previously proposed model for automatic PIANO-FINGERING on our dataset and then fine-tuning it on manually labeled piano fingering data, we achieve state-of-the-art results. In addition to the fingering extraction method, we also introduce a novel method for transferring deep-learning computer-vision models to work on out-of-domain data, by fine-tuning it on out-of-domain augmentation proposed by a Generative Adversarial Network (GAN).

For demonstration, we anonymously release a visualization of the output of our process for a single video on `https://youtu.be/Gfs1UWQhr5Q`

## 1 INTRODUCTION

Learning to play the piano is a hard task taking years to master. One of the challenging aspects when learning a new piece is the fingering choice in which to play each note. While beginner booklets contain many fingering suggestions, advanced pieces often contain none or a select few. Automatic prediction of PIANO-FINGERING can be a useful addition to new piano learners, to ease the learning process of new pieces. As manually labeling fingering for different sheet music is an exhausting and expensive task[1], In practice previous work (Parncutt et al., 1997; Hart et al., 2000; Jacobs, 2001; Kasimi et al., 2007; Nakamura et al., 2019) used very few tagged pieces for evaluation, with minimal or no training data.

In this paper, we propose an automatic, low-cost method for detecting PIANO-FINGERING from piano playing performances captured on videos which allows training modern - data-hungry - neural networks. We introduce a novel pipeline that adapts and combines several deep learning methods which lead to an automatic labeled PIANO-FINGERING dataset. Our method can serve two purposes: (1) an automatic "transcript" method that detects PIANO-FINGERING from video and MIDI files, when these are available, and (2) serve as a dataset for training models and then generalize to new pieces.

Given a video and a MIDI file, our system produces a probability distribution over the fingers for each played. Running this process on large corpora of piano pieces played by different artists, yields a total of 90 automatically finger-tagged pieces (containing 155,107 notes in total) and results in the first public large scale PIANO-FINGERING dataset, which we name APFD. This dataset will grow over time, as more videos are uploaded to YouTube. We provide empirical evidence that APFD is valuable, both by evaluating a model trained on it over manually labeled videos, as well as its usefulness by fine-tuning the model on a manually created dataset, which achieves state-of-the-art results.

The process of extracting PIANO-FINGERING from videos alone is a hard task as it needs to detect keyboard presses, which are often subtle even for the human eye. We, therefore, turn to MIDI files to obtain this information. The extraction steps are as follows: We begin by locating the keyboard and identify each key on the keyboard (§3.2). Then, we identify the playing hands on top of the keyboard

---

[1]Nakamura et al. (2019) privately reported labeling time of 3-12 seconds per note.

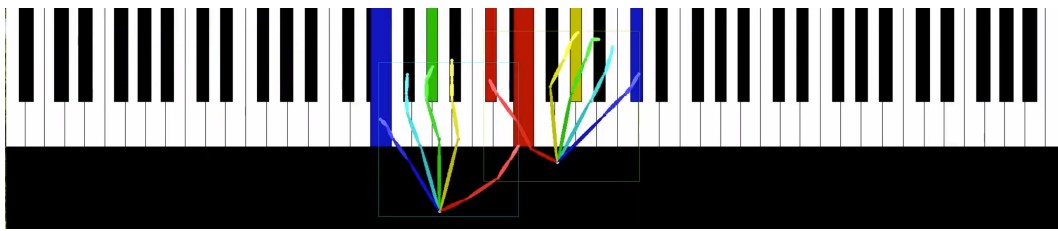

(a) A recreation of the piano, the hand pose estimation and the predicted fingering, which are colored in corresponding colors to the fingers.

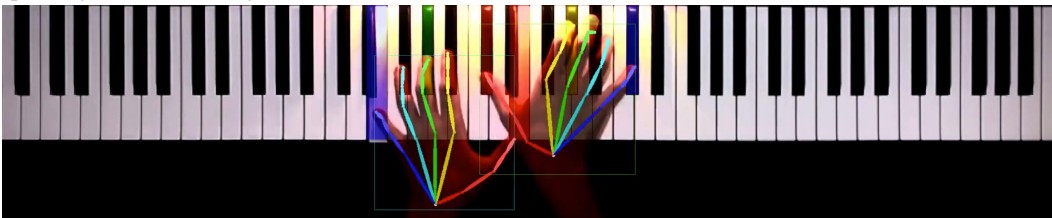

(b) Overlay of what we perceive on the original frame.

Figure 1: Example for detection of the overall system. We manage to automatically detect what fingers pressed each keynote of the piano.

(§3.3), and detect the fingers given the hands bounding boxes (§3.4). Next, we align between the MIDI file and its corresponding video (§3.6) and finally assign for every pressed note, the finger which was most likely used to play it (§3.5). Albeit the expectation from steps like hand detection and pose estimation, which were extensively studied in the computer-vision literature, we find that in practice, state-of-the-art models do not excel in these tasks for our scenario. We therefore address these weaknesses by fine-tuning an object detection model §3.3 on a new dataset we introduce and train a CycleGAN (Zhu et al., 2017) to address the different lighting scenarios with the pose estimation model §3.4.

## 2 BACKGROUND

PIANO-FINGERING was previously studied in multiple disciplines, such as music theory and computer animation (Parncutt et al., 1997; Hart et al., 2000; Jacobs, 2001; Kasimi et al., 2007; Zhu et al., 2013; Nakamura et al., 2019).

The fingering prediction task is formalized as follows: Given a sequence of notes, associate each note with a finger from the set $\{1, 2, 3, 4, 5\} \times \{L, R\}$. This is subject to constraints such as the positions of each hand, anatomical plausibility of transitioning between two fingers, the hands' size, etc. Each fingering sequence has a cost, which is derived from a transition cost between two fingers.

Early work modeled fingering prediction as a search problem, where the objective is to find the optimal fingering sequence to play all the notes with. A naive approach to finding the best sequence is to exhaustively evaluate all possible transitions between one note to another which is not computationally feasible. By defining a transition matrix corresponding to the probability or "difficulty" of transitioning from one note to another - one can calculate a cost function, which defines the predicted sequence likelihood. Using a search algorithm on top of the transitions allows finding a globally optimal solution. This solution is not practical as well, due to the exponential complexity, and therefore heuristics or pruning are employed to reduce the space complexity. The transition matrix can be manually defined by heuristics or personal estimation (Parncutt et al., 1997; Hart et al., 2000), or instead, not relying on a pre-defined set of rules, and use a Hidden Markov Model (HMM) to learn the transitions (Yonebayashi et al., 2007; Nakamura et al., 2014). In practice, (Yonebayashi et al., 2007) leaves the parameter learning to future work, and instead they manually fine-tune the transition matrix.

On top of the transition matrix, practitioners suggested using dynamic programming algorithms to solve the search (Hart et al., 2000). Another option to solve the huge search space is to use a

search algorithm such as Tabu search (Glover & Laguna, 1998) or variable neighborhood search (Mladenović & Hansen, 1997), to find a global plausible solution (Balliauw et al., 2015; 2017). These works are either limited by the defined transition rules, or by making different assumptions to facilitate the search space. Such assumptions come in the form of limiting the predictions to a single hand, limiting the evaluation pieces to contain no chords, rests or substantial lengths during which player can take their hand off the keyboard. Furthermore, all of these works have very small evaluation sets, which in practice makes it hard to compare different approaches, and does not allow to use more advanced models such as neural networks.

In this work, we continue the transition of search-based methods that optimize a set of constraints with learning methods that try to imitate human behavior by the use of large datasets. In practice, these methods require lots of training data to achieve good performance, and fingering labeling is particularly expensive and hard to obtain. One way to automatically gather rich fingering data with full hand pose estimation is by using motion capture (MOCAP) gloves when playing the piano. Zhu et al. (2013) suggests a rule-based and data-based hybrid method, initially estimating fingering decisions using a Directed Acyclic Graph (DAG) based on rule-based comfort constraints which are smoothed using data recorded from limited playing sessions with motion capture gloves. As MOCAP requires special equipment and may affect the comfort of the player, other work, (Takegawa et al., 2006) tried to automatically detect piano fingering from video and MIDI files. The pianist's fingernails were laid with colorful markers, which were detected by a computer vision program. As some occlusions can occur, they used some rules to correct the detected fingering. In practice, they implemented the system with a camera capturing only 2 octaves (out of 8) and performed a very limited evaluation. The rules they used are simple (such as: restricting one finger per played note, two successive notes cannot be played with the same finger), but far from capturing real-world scenarios.

Previous methods for automatically collecting data (Takegawa et al., 2006; Zhu et al., 2013) were costly, as apart of the equipment needed during the piece playing, and the data-collectors had to pay the participating pianists. In our work, we rely solely on videos from YouTube, meaning costs remain minimal with the ability to scale up to new videos.

Recently, Nakamura et al. (2019) released a relatively large dataset of manually labeled PIANO-FINGERING by one to six annotators, consisting of 150 pieces, with partially annotated scores (324 notes per piece on average) with a total of 48,726 notes matched with 100,044 tags from multiple annotators. This is the largest annotated PIANO-FINGERING corpus to date and a valuable resource for this field. The authors propose multiple methods for modeling the task of PIANO-FINGERING, including HMMs and neural networks, and report the best performance with an HMM-based model. In this work, we use their dataset as a gold dataset for comparison and adapt their model to compare to our automatically generated dataset.

## 3    OUR APPROACH: EXTRACTING FINGERING FROM ONLINE VIDEOS

There is a genre of online videos in which people upload piano performances where both the piano and the hands are visible. On some channels, people not only include the video but also the MIDI file recorded while playing the piece. We propose to use machine learning techniques to extract fingering information from such videos, enabling the creation of a large dataset of pieces and their fingering information. This requires the orchestration and adaptation of several techniques, which we describe below.

The final output we produce is demonstrated in Figure 1, where we colored both the fingers and the played notes based on the pose-estimation model (§3.4) and the predicted fingers that played them (§3.5). Note that the ring fingers of both hands as well as the index finger of the left hand and the middle finger of the right hand do not press any note in this particular frame, but may play a note in others. We get the information of played notes from the MIDI events.

### 3.1    DATA SOURCE

We extract videos from `youtube.com`, played by different piano players on a specific channel containing both video and MIDI files. In these videos, the piano is filmed in a horizontal angle

directly to the keyboard, from which both the keyboard and hands are displayed (as can be seen in Figure 1).

**MIDI files** A standard protocol for the interchange of musical information between musical instruments, synthesizers, and computers Musical Instrument Digital Interface (MIDI) is a standard format for the interchange of musical information between electronic musical instruments. It consists of a sequence of events describing actions to carry out, when, and allows for additional attributes. In the setup of piano recording, it records what note was played in what time for how long and its pressure strength (velocity). We only use videos that come along with a MIDI file, and use it as the source for the played notes and their timestamp.

## 3.2 Keyboard and Boundaries Detection

To allow a correct fingering assignment, we first have to find the keyboard and the bounding boxes of the keys. We detect the keyboard as the largest continuous bright area in the video and identify key boundaries using standard image processing techniques, taking into account the expected number of keys and their predictable location and clear boundaries. For robustness and in order to handle the interfering hands that periodically hide parts of the piano, we combine information from multiple random frames by averaging the predictions from each frame.

## 3.3 Hand Detection

A straightforward approach for getting fingers locations in an image is to use a pose estimation model directly on the entire image. In practice, common methods for full-body pose estimation such as OpenPose (Cao et al., 2017) containing hand pose estimation (Simon et al., 2017), make assumptions about the wrist and elbow locations to automatically approximate the hands' locations. In the case of piano playing, the elbow does not appear in the videos, therefore these systems don't work. We instead, turn to a pipeline approach where we start by detecting the hands, cropping them, and passing the cropped frames to a pose estimation model that expects the hand to be in the middle of the frame.

Object Detection (Viola & Jones, 2001; Redmon et al., 2016; Lin et al., 2017a;b), and specifically Hand Detection (Simon et al., 2017; Kölsch & Turk, 2004; Sridhar et al., 2015) are well studied subjects. However, out of the published work providing source code, the code was either complicated to run (e.g. versioning, mobile-only builds, supporting specific GPU, etc.), containing private datasets, or only detecting hands with no distinction between left and right, which is important in our case.

We, therefore, created a small dataset with random frames from different videos, corresponding to 476 hands in total evenly split between left and right[2]. We then fine-tuned a pre-trained object detection model (Inception v2 (Ioffe & Szegedy, 2015), based on Faster R-CNN (Ren et al., 2015), trained on COCO challenge (Lin et al., 2014)) on our new dataset. The fine-tuned model works reasonably well and some hand detection bounding boxes are presented in Figure 1. We release this new dataset and the trained model alongside the rest of the resources developed in this work.

Having a working hand-detection model, we perform hand detection on every frame in the video. If more than two hands are detected, we take the 2 highest probability defections as the correct bounding boxes. If two hands are detected with the same label ("left-left" or "right-right"), we discard the model's label, and instead choose the leftmost bounding box to have the label "left" and the other to have the label "right" - which is the most common position of hands on the piano.

## 3.4 Finger Pose Estimation

Having the bounding box of each hand is not enough, as in order to assign fingers to notes we need the hand's pose. How can we detect fingers that pressed the different keys? We turn to pose estimation models, a well-studied subject in computer vision and use standard models (Wei et al., 2016).

Using off-the-shelve pose estimation model, turned to often fail in our scenario. Some failure example are presented in Figure 2c where the first pose is estimated correctly, but the rest either have

---

[2]The data was labeled by the first author using *labelImg*.

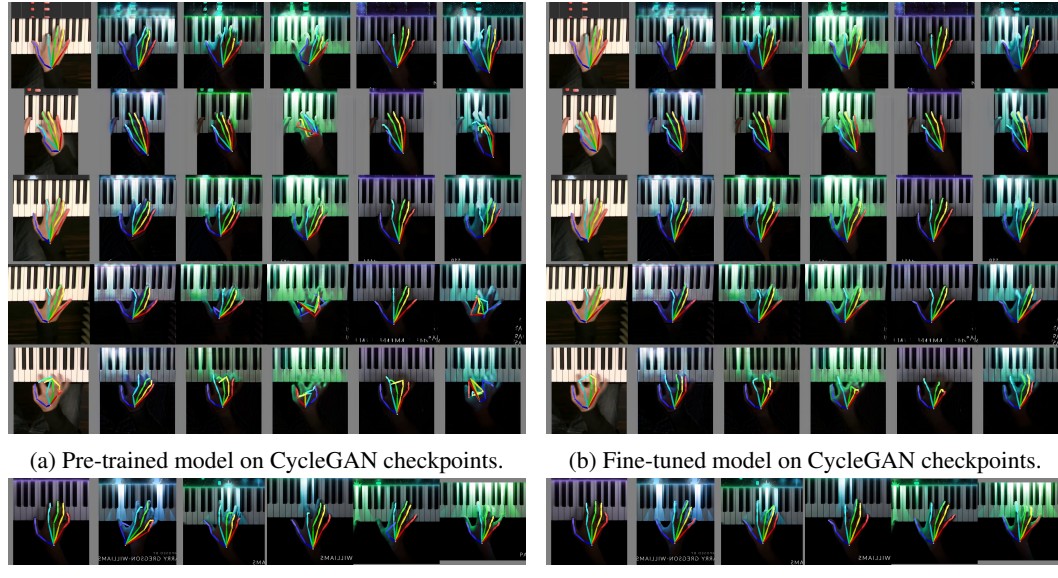

(a) Pre-trained model on CycleGAN checkpoints.      (b) Fine-tuned model on CycleGAN checkpoints.

(c) Pre-trained model on real-world data.      (d) Fine-tuned model on real-world data.

Figure 2: Output of CycleGAN on 5 images, with 5 different checkpoints. The colors drawn on the hands depict the pose estimation of the hands.

wrong finger positions, shorter, or broken fingers. The videos we use contain visual effects (e.g. LED lights are turned on every time a key is pressed), and such the pose-estimation models exhibit two failures: (1) when the LED colors are warm (red, orange, yellow), the model sees the light as an extension to the finger, and as such poorly estimates where the keypoints are for each finger; (2) when the LED colors are cool (green, blue, white), the over-saturation seems to overwhelm the model, and it mistakenly estimates the hand's pose as much shorter, considering the lit parts of the piano as part of the background. Some examples of these errors are presented in Figure 2c. Furthermore, the videos are usually very dark, high-contrast, and blurry due to motion blur, which standard datasets (Tompson et al., 2014; Yuan et al., 2017; Simon et al., 2017) and models trained on top of them rarely encounter.

Given the observation that pose-estimation works well on well lid-images, how can we augment the pose estimation model to other lighting situations? This scenario is similar to sim2real works (Sadeghi et al., 2018; Gupta & Booher; Gamrian & Goldberg, 2018) where one wants to transfer a model from simulations to the real-world. These works learn a mapping function $G_1 : T \rightarrow S$ that transfer instances $x_i$ from the target domain $T$ (the real-world) into the source domain $S$ (the simulation), where the mapping is usually achieved by employing a CycleGan (Zhu et al., 2017). Then, models which are trained on the source domain are used on the transformation of the target domain $G_1(x_i)$ and manage to generalize on the target domain. In our setup, we seek a mapping $G_2 : S \rightarrow T$ that transforms the source domain (i.e the well-lid videos) into the target data (i.e the challenging lighted scenarios). After obtaining the transformation function $G_2$, we employ the pose estimation model $f$ on the source domain, use the transformation separately, and align the prediction to the new representation. This novel setup benefits from performance boost as we only use the transformation function offline, before training and avoid using it for every prediction. We also benefit of better generalization as we keep good performance on the source domain, and gain major performance on the target domain.

We manually detect videos and assign them into their group (well-lit or poorly-lit). Then, we automatically detect and crop hands from random frames, resulting in 21,747 well-lit hands, and 12,832 poorly lit hands. We then trained a CycleGAN for multiple epochs and chose 15 training checkpoints that produced different lighting scenarios (some examples can be seen in Figure 2). We then fine-tune a pose-estimation model on the original, well lit frames, and on the 15 transformed frames. This procedure results in a model that is robust to different lighting scenarios, as we show in Figures 2b and 2d, demonstrating its performance on different lighting scenarios.

### 3.5 PRESSED FINGER ESTIMATION

Given that we know which notes were pressed in any given frame (see §3.6 below), there is still uncertainty as to which finger pressed them. This uncertainty either comes from imperfect pose estimation, or multiple fingers located on top of a single note. We model the finger press estimation by calculating the probability of a specific finger to have been used, given the hand's pose and the pressed note:

$$argmax_{i,j}P(f_i, h_j|n_k)$$

where $i \in [1, 5]$ for the 5 fingers, $h_j \in \{h_l, h_r\}$ stands for the hand being used (left or right) and $n_k \in [1, 88]$ corresponding to the played key.

We chose to estimate the pressed fingers as a Gaussian distribution $\mathcal{N}(\mu, \sigma^2)$, where $\mu$ and $\sigma$ are defined as follows:

$$\sigma(n_k) = X_{n_{k+1}} - X_{n_k}, \quad \mu(n_k) = X_{n_k} + 0.5 * \sigma$$

$\mu$ is the center of the key on the $x$ axis and $\sigma$ is its width. The score of each finger given a note in a specific frame is defined as:

$$g(f_i, h_j|n_k, frame) = f(X_{f_i, h_j|frame}|\mu(n_k), \sigma(n_k)^2)$$

The probability of a given finger to have played a note given a note and a frame: (normalizing $g$ for all fingers)

$$p(f_i, h_j|n_k, frame) = \frac{g(f_i, h_j|n_k, frame)}{\Sigma_{n=1}^5 \Sigma_{m \in \{l,r\}} g(f_n, h_m|n_k, frame)}$$

As most keyboard presses last more than one frame, we make use of multiple frames to overcome some of the errors from previous steps and to estimate a more accurate prediction. For this reason, we aggregate the frames that were used during a key press. We treat the first frame as the main signal point, and assign each successive frame an exponentially declining weight

$$p(f_i, h_j|n_k, frame_{k_1}, ...frame_{k_n}) = \frac{\Sigma_{l=1}^n 0.5^l * p(f_i, h_j|n_k, frame_{k_l})}{\Sigma_{l=1}^n 0.5^l}$$

As finger changes can occur in later frames. Finally, we normalize the weighted sum of probabilities to achieve a probability distribution for all frames.

In our dataset, we release all probabilities for each played note, along with the maximum likelihood finger estimation. We define the "confidence" score of the extraction from a single piece, as the product of the highest probability for a finger for each note. Figure 3 shows the precision and recall of the predictions based on a cutoff for the highest probability of the note. We see a positive correlation between confidence threshold and precision, and a negative correlation between confidence and recall, meaning we can get relatively high precision for a small number of notes, or relatively low precision for a high number of notes.

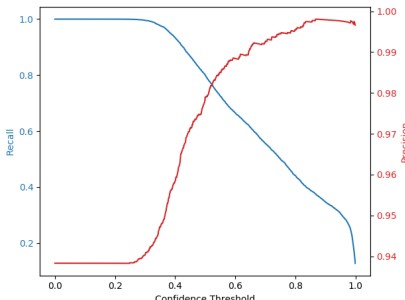

Figure 3: Precision and recall based on confidence for all 6,894 manually tagged notes

### 3.6 VIDEO AND MIDI ALIGNMENT

We consider the MIDI and video files to be complementary, as they were recorded simultaneously. The MIDI files are the source to which keys were pressed, in what time, and for how long. The videos are the source for the piano, hands, and fingers locations. These two sources are not synchronized, but as they depict the same piano performance, a perfect alignment exist (up to the video frequency resolution).

We extract the audio track from the video, and treat the first audio peak as the beginning of the piece, clipping the prior part of the video and aligning it with the first event from the MIDI file. In practice,

we determine this starting point as the first point in the signal where the signal amplitude is higher than a fifth of the mean absolute value of the entire signal.

This heuristic achieves a reasonable alignment, but we observe some alignment mismatch of 80-200ms. We tackle the misalignment by using a signal from the final system confidence (Section 3.5), where every piece gets a final score estimation after running the whole process, depicting the system confidence on the predicted notes. We look for an alignment that maximizes the system confidence over the entire piece:

$$alignment(MIDI, Video) = argmax_i score(MIDI_{t_0}, Video_{t_i})$$

where $MIDI_{t_0}$ is the starting time of the MIDI file, and $Video_{t_j}$ is the alignment time of the video. $Video_{t_0}$ is obtained by the heuristic alignment described in the previous paragraph. We use the confidence score as a proxy of the alignment precision and search the alignment that maximizes the confidence score of the system. More specifically, given the initial offset from the audio-MIDI alignment, we take a window of 1 second in frames (usually 25) for each side and compute the score of the final system on the entire piece. We choose the offset that results in the best confidence score as the alignment offset.

### 3.7 THE RESULTING DATASET: APFD

We follow the methods described in this section, and use it to label 90 piano pieces from 42 different composers with 155,107 notes in total. On average, each piece contains 1,723 notes.

## 4 RESULTS

In this section, we present multiple evaluations of our overall system. We begin by evaluating the entire process where we assess how the overall system performs on predicting the pressed fingering. Next, we use the dataset we collected and train a PIANO-FINGERING model. We fine-tune this model on a previously manually-annotated dataset for PIANO-FINGERING and show using our data achieves better performance.

As piano pieces are usually played by two hands, we avoid modeling each hand separately, and instead use their symmetry property, and simply flip one hand's notes, matching it to the piano scale, following previous work practice (Nakamura et al., 2019).

For evaluation, we use the match rate between the prediction and the ground truth. For cases where there is a single ground truth, this is equivalent to accuracy measurement. When more than one labeling is available, we simply average the accuracies with each labeling.[3]

### 4.1 FINGER PRESS ESTIMATION EVALUATION

As the pose estimation is one of the major components directly affecting our system's performance, we isolate this part in order to estimate the gain from fine-tuning using the CycleGAN (Section 3.4).

We manually annotated five random pieces from our dataset by marking the pressing finger for each played note in the video. Then, by using our system (§3.5), we estimate what finger was used for each key. We use the confidence score produced by the model as a threshold to use or discard the key prediction of the model, and report precision, recall and F1 scores of multiple thresholds. Moreover, we compare the scores of these pieces with and without using the CycleGAN. We do so for the five annotated pieces and report these results in Table 1. When considering a high confidence score (>90%) both the pre-trained and fine-tuned models correctly mark all considered notes (which consist of between 34-36% of the data). However, when considering decreasing confidences, the fine-tuned model manages to achieve higher precision and higher recall, contributing to an overall higher f1 score. With no confidence threshold (i.e using all fingering predictions), the pre-trained model achieves 93% F1, while the fine-tuned one achieves 97% F1, a 57% error reduction.

### 4.2 AUTOMATIC PIANO FINGERING PREDICTION

---

[3]This matches the *general match rate* evaluation metric in (Nakamura et al., 2019).

| Confidence | c > 0.9 | | | c > 0.75 | | | c > 0.5 | | | c > 0 | | |
|---|---|---|---|---|---|---|---|---|---|---|---|---|
| Metrics | p | r | f1 | p | r | f1 | p | r | f1 | p | r | f1 |
| Pretrained | 1.0 | 0.34 | 0.51 | 0.99 | 0.49 | 0.65 | 0.95 | 0.78 | 0.86 | 0.88 | 1.0 | 0.93 |
| FT Single Frame | 0.99 | 0.36 | 0.53 | 0.99 | 0.52 | 0.68 | 0.97 | 0.8 | 0.88 | 0.9 | 1.0 | 0.95 |
| FT Multiple Frames | 1.0 | 0.35 | 0.52 | 0.99 | 0.5 | 0.67 | 0.98 | 0.8 | 0.88 | 0.94 | 1.0 | 0.97 |

Table 1: Precision, Recall, F1 scores for 6,894 notes we manually tagged across 5 different pieces, at different confidence points. The first row depict the performance of the system with the vanilla pose estimation. The second and the third rows present the results on the fine-tuned (FT) pose estimation model 3.4 on a single and multiple frames accordingly.

In order to assess the value of APFD, we seek to show its usefulness on the end task: Automatic Piano Fingering. To this end, we train a standard sequence tagging neural model using our dataset, evaluating on the subset of the videos we manually annotated. Then, we fine-tune this model on PIG (Nakamura et al., 2019), a manually labeled dataset, on which we achieve superior results than simply training on that dataset alone.

| Training/Dataset | APFD | PIG |
|---|---|---|
| Previous Neural | — | 61.3 |
| Previous SOTA | — | 64.5 |
| Human Agreement | — | 71.4 |
| PIG | 49.9 | 64.1 |
| + Fine-tune APFD | **73.6** | 54.4 |
| APFD | 73.2 | 55.2 |
| + Fine-tune PIG | 62.9 | **66.8** |

Table 2: Results of training and fine tuning on different dataset.

We model the PIANO-FINGERING as a sequence labeling task, where given a sequence of notes $n_1, n_2, ..., n_n$ we need to predict a sequence of fingering: $y_1, y_2, ..., y_n$, where $y_i \in \{1, 2, 3, 4, 5\}$ corresponding to 5 fingers of one hand. We employ a standard sequence tagging technique, by embedding each note and using a BiLSTM on top of it. On every contextualized note we then use a Multi-Layer-Perceptron (MLP) to predict the label. The model is trained to minimize cross-entropy loss. This is the same model used in (Nakamura et al., 2019), referred to as DNN (LSTM).

### 4.2.1 PIANO FINGERING MODEL

Nakamura et al. (2019) didn't use a development set, therefore in this work, we leave 1 piece from the training set and make it a development set. Our dataset—APFD—is composed of 90 pieces, which we split into 75/10 for training and development sets respectively, and use the 5 manually annotated pieces as a test set. We note that the development set is silver data (automatically annotated) and probably contains mistakes. The results are summarized in Table 1. We run the same architecture by (Nakamura et al., 2019) with some different hyperparameters and achieve 71.4%/64.1% on our and PIG's test set respectively. To evaluate the usefulness of our data to PIG's data, we use the model trained on our silver data and fine-tune it on PIG. This results in 66.8% accuracy, 2.3% above the previous state-of-the-art model which was achieved by an HMM (Nakamura et al., 2019). We attribute this gain in performance to our dataset, which both increases the number of training examples and allows to train bigger neural models which excel with more training examples. We also experiment in the opposite direction and fine-tune the model trained on PIG with our data, which result in 73.6% accuracy, which is better than training on our data alone, achieving 73.2% accuracy.

## 5 DISCUSSION AND FUTURE WORK

In this work, we present an automatic method for detecting PIANO-FINGERING from MIDI and video files of a piano performance. We employ this method on a large set of videos, and create the first large scale PIANO-FINGERING dataset, containing 90 unique pieces, with 155,107 notes in total. We show this dataset—although being noisy–is valuable, by training a neural network model on it, fine-tuning on a gold dataset, where we achieve state-of-the-art results. In future work, we intend to improve the data collection by improving the pose-estimation model, better handling high speed movements and the proximity of the hands, which often cause errors in estimating their pose. Furthermore, we intend to design improved neural models that can take previous fingering predictions into account, in order to have a better global fingering transition.

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

# Appendices

## A DATASET SAMPLES

For every video our system extracts PIANO-FINGERING from it also outputs the video overlayed by the estimation of the piano keys, an indication of which notes are played, and what fingers are being used to play them (the key's color), up to two bounding boxes, for both hands (where light blue means left hand, and light green means right hand), and the pose estimation for each hand.

We include the output videos for all of the pieces we manually annotated to visually demonstrate our system and its accuracy on different playing cases. The videos were uploaded to a new, anonymous YouTube channel, and each contain a link to the original video in the description.

River Flows in You: `https://youtu.be/Gfs1UWQhr5Q`
Faded: `https://youtu.be/LU2ibOW6z7U`

Moonlight Sonata 1st Movement: `https://youtu.be/wp8j239fs9o`
Rondo Alla Turca: `https://youtu.be/KqTaPfoIuuE`
Nocturne in E Flat Major (Op. 9 No. 2): `https://youtu.be/xXHUUzTa5vU`

