# OpenReview forum: "At Your Fingertips: Automatic Piano Fingering Detection"
_ICLR.cc/2020/Conference — Reject_

### Official Review · AnonReviewer1 · 2019-10-14
**Official Blind Review #2478**

**Rating:** 1

**Review:**

<Strengths>
+ This paper addresses an interesting and practically important problem: detection of piano fingering from videos and MIDI files. Fingering is a valuable source for piano learners and has to be manually annotated otherwise. The proposed approach is automatic and low-cost from playing videos.
+ This paper collects a large-scale dataset for piano-fingering named APFD, including 90 finger-tagged pieces with 155K notes.
+ The paper reads very well.


<Weakness>
1. One major weakness of this work is lack of technical novelty.
- As described in section 3 in detail, the proposed approach consists of a sequence of well-known techniques (e.g. Faster R-CNN for hand detection and CycleGAN for finger pose estimation) and is largely based on lots of heuristics in every step of the procedure.
- Thus, the method may be practically viable but bear little technical novelty.

2. Experimental results are rather weak.
- Only a single baseline is used in the existing PIG dataset, while no baseline method is compared for the new APFD dataset,  for which more baselines may need to be implemented and compared.
- The proposed approach (64.1) is slightly worse than the previous SOTA (64.5) in the PIG dataset, although it is improved by fine-tuning with the APFD data that are not available for the previous SOTA. Thus, no experimental evidence is presented in the paper to convince that the proposed approach is better than existing ones.

<Conclusion>
Although this work is practically promising, my initial decision is ‘reject’ mainly due to lack of technical novelty and limited experiments.

**Experience Assessment:**

I have published in this field for several years.

**Review Assessment: Checking Correctness Of Derivations And Theory:**

I assessed the sensibility of the derivations and theory.

**Review Assessment: Checking Correctness Of Experiments:**

I carefully checked the experiments.

**Review Assessment: Thoroughness In Paper Reading:**

I read the paper thoroughly.

---

> ### Author Response · Authors · 2019-11-06
> **Main reply to Review #1**
>
> Thank you for your review.
>
> We would like to address your review:
> 1. We are not aware of any works using CycleGAN for pose estimation, or any works using sim2real in that manner (of fine-tuning the algorithms). It would be great to get some citations.
> Regarding the novelty of our method, here is a list of what we think are the core novel ideas of our method:
> a. We show how it is possible to segment each key of the piano.
> b. We show the failures of current pose detection models, and devise a new, general way to address such failures, that we show is robust - and for us, reduces 50% of the accumulative error.
> c. We design a new way to align between MIDI and piano video recordings, which is accurate up to the video sampling rate.
> d. We design an algorithm to assume "collisions" between the detected fingers in multiple frames, and the piano keys, from very noisy data.
>
> 2. While we agree more baselines are better to have, in our experiments we aim to convince the reader that our data is indeed valuable, and not just noise, and we think it manages to show that. We do not aim to create a new method for automatic fingering in this work.
> While indeed the proposed approach trained on the PIG dataset gets slightly worse results from the SOTA on the PIG dataset, the case we are trying to make is not that our method is better, it is that our data is good and valuable, and can be used for future developed methods.
>
>
> Again, we would like to thank you, and perhaps get your feedback on what should be improved in this paper? What experiments/analysis you would like to see?

---

### Official Review · AnonReviewer2 · 2019-10-18
**Official Blind Review #2**

**Rating:** 3

**Review:**

The paper is a nice piece of works which clearly articulates the objective and the subsequent discussion. The focus of the paper--i.e. disclose the difficulties of piano fingering data annotation and the proposal of automating this process by automatically extracting fingerings from public videos and MIDI files, using  computer-vision DNN-based algorithms —although not really mainstream, it does provide some practical insights using a couple of experimental settings (piano fingering model and prediction) to help the readers.

I really enjoyed reading this paper. I think that it can be considered a relevant and interesting piece of work, very well written and clear. Furthermore, providing new benchmarks/datasets/competitions for the AI community is always refreshing. Also, the results seem believable and solid, and potentially useful.

My only concern is that, although the rationale and utility of the paper is clear, the rest of the paper is somewhat incremental/engineering piece which depends somehow on previous works (see Nakamura,2019). I fail to see much novel scientific contribution to the area of research (apart from the dataset) and I’m not sure whether there are enough scientific technical advancements. Furthermore, the experimental setting is somewhat limited, and it is not clear whether results are statistically significant.


**Experience Assessment:**

I have read many papers in this area.

**Review Assessment: Checking Correctness Of Derivations And Theory:**

N/A

**Review Assessment: Checking Correctness Of Experiments:**

N/A

**Review Assessment: Thoroughness In Paper Reading:**

N/A

---

> ### Author Response · Authors · 2019-11-06
> **Main reply to Review #2**
>
> Thank you for your review.
>
> We would like to address your review:
> - **... the rest of the paper is somewhat incremental/engineering piece ...** - while true, that engineering was involved, we think that in most steps of our pipeline we introduce new, novel methods / usages to tackle different issues. (described further in the rest of this comment).
> - **... which depends somehow on previous works (see Nakamura,2019)** - We don't see what you are referring to, as in their work they release a manually annotated dataset, while in this work we devise an unsupervised method to extract such annotations from videos and MIDI files.
> - **I fail to see much novel scientific contribution to the area of research (apart from the dataset) and I’m not sure whether there are enough scientific technical advancements.** - I will now list the novel parts of this work, that were used in order to get a good dataset:
> 1. We show how it is possible to segment each key of the piano.
> 2. We show the failures of current pose detection models, and devise a new, general way to address such failures, that we show is robust - and for us, reduces 50% of the accumulative error.
> 3. We design a new way to align between MIDI and piano video recordings, which is accurate up to the video sampling rate.
> 4. We design an algorithm to assume "collisions" between the detected fingers in multiple frames, and the piano keys, from very noisy data.
>
> - **Furthermore, the experimental setting is somewhat limited, and it is not clear whether results are statistically significant.** - You are correct that we did not perform a statistical significance test, we will do such for the final version of the paper. We show 50% error reduction between previous SOTA and human agreement results. The experimental is not meant to devise a new method for piano fingering, instead, it is design to convince the reader that our data is indeed valuable, and not just noise, and we think it manages to show that.
>
> Again, we would like to thank you, and perhaps get your feedback on what should be improved in this paper? What experiments/analysis you would like to see?

---

### Official Review · AnonReviewer3 · 2019-10-22
**Official Blind Review #3**

**Rating:** 1

**Review:**

In this paper, the authors proposed an automatic piano fingering algorithm, that accepts YouTube videos and corresponding MIDI files and outputs fingering prediction for each note. The claimed contribution is two-fold: First, they proposed the algorithm, and second, they claim that the algorithm can be used to automatically generate large datasets for piano fingering problems. The motivation is clearly stated and convincing. The overall algorithm is mainly described.

However, I would like to reject this paper. Major issues:

* Some key information is missing in Section 3.6, which is the only section that shows technical details: What is X_{n_k}? How is that related to the estimated finger poses? What is the function f in the definition of function g? (Also, it would be helpful to label the equations for clarification.) Are you doing Bayesian inference? With the key information missing, it is hard to fully understand the remaining technical details in this section.
* Their experimental results cannot properly support their claims. In Section 4.2, the authors try to show the strength of their proposed piano fingering algorithm by comparing their automatically annotated dataset APFD with an existing manually annotated dataset PIG. The authors showed the evaluation results of models trained and fine-tuned with different datasets. However, this is not an acceptable comparison for me, due to several reasons.
First, in order to show the strength of automatic piano fingering prediction, it is much better to directly run the prediction algorithm on datasets with known labels. According to the related work section, there is at least one existing work by Takegawa et al. that uses videos and MIDI files to detect piano fingering. Can you compare your algorithm with theirs?
Second, it is essentially unreliable to compare two datasets by comparing the performance of two prediction models, as there are too many implementation details that are almost impossible to control.
Third, it is not clear how we should compare the testing errors in Table 2. Yes, a model initially trained on PIG and fine-tuned on APFD may perform better than a model trained merely on APFD, but does that suggest anything (and the advantage is just 0.4%)? Similarly, the experimental result that an MLP model initially trained on APFD then fine-tuned with PIG works better than an HMM model that is trained with PIG data alone cannot prove anything. There are too many possible reasons that may lead to this experimental result.
* How is this method more attractable than the existing ones? There are neither experimental comparisons nor high-level justifications of why the existing algorithms are not applicable to the given scenario. In Section 2, although the authors described a good number of existing work on piano-fingering and their drawbacks, they failed to point out the strength of their paper as a comparison. As a result, the strength of this paper is still unclear after reading this section. How does this paper avoid the drawbacks of these previous papers?
* The writing of this paper needs to be greatly improved. It takes a lot of effort to literally understand this paper: There may be missing parts, misplaced clauses, and broken logic between sentences. I have listed several examples in the minor issues part.


Minor issues:
* In the first paragraph of Section 1: The sentence before 'In practice ...' is incomplete.
* In the last paragraph of Section 1: Missing brackets for \textsection 3.3 and \textsection 3.4. Also, 'on A new dataset we introduce' should be 'on THE new dataset we introduce'.
* On page 3, the sentence 'In this work, we continue the transition of search-based methods that optimize a set of constraints with learning methods that ...' is not making sense to me. Do you mean that your work is an extension of search-based methods, or do you mean that your work is not a search-based method? Also, are you optimizing a set of constraints, or optimizing with a set of constraints?
* On page 3, the last sentence in Section 2: '... and adapt their model to compare TO our ...' should be '... and adapt their model to compare WITH our ...'. The last part of this sentence is also a bit confusing: How do you compare a model with a dataset?
* On page 4, the paragraph starting with 'MIDI files': The first two sentences are almost the same; the period between them is missing. I guess one of them should be deleted. The following sentences in this paragraph are also subject to grammatical errors. For example, the sentence 'It consists of a sequence of events ... to carry out, WHEN, and allows for ...' is not a complete sentence. 'We only use videos that come along with a MIDI file' -> 'We only use videos that come along with MIDI files'.
* On page 5, last paragraph in Section 3.3: 'highest probability defections' -> 'highest probability detections'.
* The last paragraph on Page 5: 'Using off-the-shelve ...' -> 'Use off-the-shelf ...'.
* In Section 4.2.1, the corresponding result is Table 2, instead of Table 1.

**Experience Assessment:**

I do not know much about this area.

**Review Assessment: Checking Correctness Of Derivations And Theory:**

N/A

**Review Assessment: Checking Correctness Of Experiments:**

I carefully checked the experiments.

**Review Assessment: Thoroughness In Paper Reading:**

I read the paper thoroughly.

---

### Decision · Program_Chairs · 2019-12-19

**Decision:**

Reject

**Comment:**

The paper shows an automatic piano fingering algorithm. The idea is good. But the reviewers find that the novelty is limited and it is an incremental work. All the reivewers agree to reject.